# The Pathogenic Role of Interferons in the Hyperinflammatory Response on Adult-Onset Still’s Disease and Macrophage Activation Syndrome: Paving the Way towards New Therapeutic Targets

**DOI:** 10.3390/jcm10061164

**Published:** 2021-03-10

**Authors:** Ilenia Di Cola, Piero Ruscitti, Roberto Giacomelli, Paola Cipriani

**Affiliations:** 1Department of Biotechnological and Applied Clinical Sciences, University of L’Aquila, 67100 L’Aquila, Italy; ilenia.dicola@graduate.univaq.it (I.D.C.); paola.cipriani@univaq.it (P.C.); 2Rheumatology and Immunology Unit, Department of Medicine, University of Rome Campus Biomedico, 00128 Rome, Italy; r.giacomelli@unicampus.it

**Keywords:** adult-onset Still’s disease, macrophage activation syndrome, IFN-γ

## Abstract

Adult-onset Still’s disease (AOSD) is a systemic inflammatory disorder of unknown aetiology affecting young adults, which is burdened by life-threatening complications, mostly macrophage activation syndrome (MAS). Interferons (IFNs) are signalling molecules that mediate a variety of biological functions from defence against viral infections, to antitumor and immunomodulatory effects. These molecules have been classified into three major types: IFN I, IFN II, IFN III, presenting specific characteristics and functions. In this work, we reviewed the role of IFNs on AOSD and MAS, focusing on their pathogenic role in promoting the hyperinflammatory response and as new possible therapeutic targets. In fact, both preclinical and clinical observations suggested that these molecules could promote the hyperinflammatory response in MAS during AOSD. Furthermore, the positive results of inhibiting IFN-γ in primary hemophagocytic lymphohistiocytosis may provide a solid rationale to arrange further clinical studies, paving the way for reducing the high mortality rate in MAS during AOSD.

## 1. Introduction

Adult-onset Still’s disease (AOSD) is an inflammatory disease usually affecting young adults [1]. AOSD is associated with a very heterogeneous clinical picture, a triad of high fever, arthritis, and evanescent pink salmon skin rash are commonly observed [2]. Furthermore, a multiorgan involvement of the disease is recognised, including liver involvement, splenomegaly, and poly-serositis [2]. A typical hyperferritinemia is observed in these patients, associated with increases of C-reactive protein (CRP) and erythrocyte sedimentation rate (ESR) [1]. Additionally, patients with AOSD experience life-threatening complications, which may rapidly evolve into multiple-organ failure and death [3]. These patients would frequently develop macrophage activation syndrome (MAS), a secondary form of hemophagocytic lymphohistiocytosis (HLH) [4,5]. The latter is characterised by continuous high fever, extreme hyperferritinemia, pancytopenia, and histopathological evidence of hemophagocytosis by activated macrophages, typically in bone marrow [5,6]. 

Although it is typical, this histological finding is not mandatory for HLH diagnosis since it cannot be recognized at the beginning of the disease in bone marrow biopsies [4]. Another important characterisation of HLH is the organomegaly, splenomegaly, and hepatomegaly frequently recognized in these patients [4]. In addition, it was proposed that AOSD and MAS may be considered part of the same disease spectrum, sharing clinical and pathogenic features, and in which AOSD may represent a milder form [7]. Furthermore, these diseases have been recently included in the so-called hyperferritinemic syndrome, which, together with catastrophic anti-phospholipid syndrome and septic shock, share similar clinical and laboratory features, including very high levels of ferritin [8]. 

As far as the pathogenesis is concerned, AOSD is considered at the crossroads between auto-inflammatory and autoimmune diseases [9]. Both the innate and adaptative arms of the immune system are called upon in the pathogenic mechanisms underlying this disease [10]. The pathogenic mechanisms of MAS have not been fully clarified yet, but recently a multi-layer pathogenic model was proposed [6]. Both genetic predisposition and several triggers may contribute to the development of a cytolytic dysfunction, prolonging the survival of target cells and enhancing antigen presentation to overproduce proinflammatory cytokines, leading to full-blown MAS syndrome [5,6,11]. In this context, the role of interferons (IFNs) was pointed out mainly for inducing cytokine storm syndrome and MAS occurrence during AOSD [5,6,11]. On these bases, in this work we reviewed the role of IFNs on AOSD and MAS, focusing on their pathogenic role in promoting the hyperinflammatory response and as new possible therapeutic targets. 

## 2. Interferons

In 1957, a molecule was first described with the ability to “interfere” with viral replication and protect cells from infection, which was called an IFN [12]. Since then, a growing body of evidence has shown that multiple IFNs exist which mediate a variety of biological functions from defence against viral infections to antitumor and immunomodulatory effects [13]. IFNs are classified into three main groups according to chromosomal location, their aminoacidic sequence, and specific receptors: i. type I IFNs (-α, -β, -δ, -ε, -ζ, -κ, -τ, and -ω); ii. type II IFN (-γ); iii. type III IFNs (-λ1, -λ2, -λ3). Type I IFNs and IFN-γ are physiologically expressed and are increased by stress and infections [13]. IFNs are critical effectors of both innate and adaptive immune responses, associated with the development of immune cell populations and their activation in response to pathogens, cancers, and other conditions [14]. In addition, the elevated production of IFNs is recognised during both autoimmune and autoinflammatory diseases [15]. This increases the expression of target genes and the canonical interferon-stimulated genes (ISGs) in affected tissues and in circulating blood cells, thus defining the “IFN signature” [14]. The latter is reported to be a typical characteristic of some diseases [16].

## 3. IFN I

### 3.1. Generalities

IFN-α and IFN-β are the most studied and characterised members of this class of IFNs [16]. IFN-α is encoded by more than 20 different genes. Among these, 13 lead to a functional protein in humans and 14 in mice, whereas IFN-β is encoded by a single gene in both humans and mice [16,17]. Although IFN-α and -β may regulate an overlapping set of genes, these two cytokines differ slightly in their downstream effects and in their expression pattern [18]. Other type I subtypes (IFN- δ, -ε, -ζ, -κ, -τ, and -ω) are less-often studied [16]. Type I IFNs act on most cell types and induce an antiviral state by increasing the major histocompatibility complex expression and inducing the production of chemokines and cytokines [19,20]. Furthermore, type I IFNs boost the innate arm of the immune system by stimulating the maturation of dendritic cells and the function of natural killer cells [16]. These IFNs also enhance the adaptive response of the immune system by promoting the activation of T and B cells [14]. As a major component of the innate immune system protecting against viruses, the expression of IFN-α and IFN-β is induced by viral infection [19,20]. Type I IFNs bind to the ubiquitously expressed type I IFN receptor (IFNAR) in an autocrine and paracrine manner, modulating the expression of numerous IFN-stimulated genes (ISG) which are involved in the antiviral and anti-inflammatory responses and the pro-apoptotic and anti-proliferative activities [18].

### 3.2. Pathogenic Implications in AOSD and MAS

Multiple lines of evidence indicate that type I IFNs also exert anti-inflammatory functions [21,22,23]. These anti-inflammatory phenomena were proposed because IFN-α may reduce both interleukin (IL)-1α and IL-1β production by two main pathways [23,24]. By acting on the signal transducer and activator of transcription 1 (STAT1), type I IFNs may repress the activity of the Nucleotide Binding Domain (NBD), Leucine-Rich Repeat (LRR) containing (NLR) protein 1 (NLRP1) and NLRP3 inflammasomes, thereby suppressing caspase-1-dependent IL-1β maturation [23]. These molecules could also induce the expression of IL-10 in a STAT1-dependent manner, which in turn may reduce the abundance of the pro-IL-1α and pro-IL-1β signals via STAT3 [23]. Such inflammasome inhibition by type I IFNs may also suggest a mechanism for the observed IFN-dependent suppression of IL-18 maturation, since it would also depend on inflammasome activity [23]. Because of these anti-inflammatory functions, an impaired response of type I IFNs may be implicated in the generation of the hyperinflammatory processes [18]. Patients with more severe COVID-19, during the ongoing catastrophic pandemic by SARS-CoV-2, may provide a virally induced representative model of cytokine storm syndrome, thus suggesting similarities with the underlying pathogenic mechanisms of AOSD and MAS [25,26]. Interestingly, severe coronavirus disease 2019 (COVID-19) may display many common aspects with other disorders included in hyperferritinaemic syndrome, including continuous fever and high levels of ferritin [27]. In the context of COVID-19, Hadjadj et al. observed a distinct phenotype in severe and critical patients, associated with a highly impaired type I IFN response, associated with decreased production and reduced activity [28]. In addition, the presence of neutralizing autoantibodies against type I IFNs was supposed in the inhibition of the type I IFN response [29]. These autoantibodies against type I IFNs seemed to be clinically silent until the infection, suggesting that the small quantities of such molecules could be implicated in the onset of cytokine storm syndrome [29].

Taking these observations together, the impairment of the functions of type I IFNs or their delayed response may be implicated in the development of a cytokine storm syndrome. These pathogenic alterations could be also associated with the development of MAS during AOSD, thus providing food for thought for further mechanistic studies. In fact, limited data are available about the role of IFN I in the pathogenesis of AOSD and MAS, so far. In this setting, sera levels of both IFN-α and IFN-β were studied by enzyme-linked immunosorbent assay (ELISA) in 39 AOSD patients, both during a flare of the disease and when following therapies [30]. Levels of IFN-α were detected in only one of the AOSD patients. Instead, levels of IFN-β were found in both patients with an active flare of the disease and those following therapies, without any statistically significant difference [30]. Notably, the type I IFN response on the HLH experimental model was studied in a murine model with a specific deletion of IFNAR (IFNAR-KO) [31]. HLH was induced by stimulation with an IL-10 receptor-blocking antibody and a Toll-like receptor 9 (TLR9) agonist. When IL-10 signalling was maintained, the administration of the TLR9 agonist resulted in a milder HLH in wild-type (WT) mice, with less severe hepatitis and lack of hemophagocytosis. However, thrombocytopenia and IFN-γ were similar between the IFNAR-KO and the WT mice. Despite IFN-γ levels being comparable to those of the WT mice, the IFNAR-KO mice did not develop anaemia, suggesting that type I IFNs could be involved in leading to this feature during HLH [31]. In the same model, the simultaneous administration of both an IL-10 receptor-blocking antibody and a TLR9 agonist led to fulminant HLH. The IFNAR-KO mice had less weight loss than their WT counterparts but were comparable for thrombocytopenia, hepatitis, and splenic hemophagocytosis. Furthermore, the IFNAR-KO mice treated for fulminant HLH conditions experienced the same degree of anaemia when compared to WT mice. Taking together these findings, a complex interaction between type I and type II IFNs in the pathogenesis of TLR9-mediated HLH could be suggested [31].

## 4. IFN II

### 4.1. Generalities

The type II IFN subtype is made of a single gene product: IFN-γ [16,32,33]. Its structure is different from type I IFNs, but it is classified in this family of molecules due to its antiviral effects [16]. IFN-γ binds to the nearly ubiquitously expressed receptor (IFNGR), and signals through Janus kinase 1 (JAK1) and JAK2 to phosphorylate STAT1 [13]. IFN-γ is involved in the modulation of the immune and inflammatory responses and is predominantly produced by natural killer (NK), NKT, and activated T cells [18]. It has weaker antiviral effects than type I IFNs, but potent effects on increasing major histocompatibility complex expression, antigen presentation, and chemokine production, while suppressing cell proliferation [18]. IFN-γ would be the prototypic “macrophage-activating factor” increasing cytokine and chemokine production, phagocytosis, and the intracellular killing of microbial pathogens by macrophages [33]. Furthermore, IFN-γ boosts type 1 adaptive immunity by promoting the differentiation of type 1 helper T cells, the generation of follicular helper T cells, B cell class switching, autoantibody production, and the generation of autoimmunity-associated B cells [18]. This molecule may also have protective functions by suppressing responses mediated by type 2 helper- and IL-17-producing helper T cells, inducing specialized regulatory T cells and restraining tissue damage [18]. Moreover, IFN-γ may directly enhance antigen presentation by promoting antigen processing and by inducing the expression of major histocompatibility complex molecules [18].

In this context, the involvement of the IFN-γ pathway in the pathogenic mechanisms of HLH, either primary or secondary, was proposed [30]. Although the mechanisms leading to IFN-γ-mediated immunopathology remain to be fully clarified, many data would suggest this cytokine is a crucial mediator in HLH occurrence [30,34,35,36].

### 4.2. Pathogenic Implications, Ex Vivo Observations

The pathogenic implications of IFN-γ in HLH were studied through the evaluation of neopterin levels in HLH patients [37]. Neopterin is a marker of inflammation belonging to a group of pteridines, and it is biosynthetically derived from guanosine triphosphate. It is secreted by human monocyte-derived macrophage and dendritic cells upon stimulation with IFN-γ. On these bases, it may be considered as a surrogate marker of this cytokine. Sera neopterin levels obtained at the time of diagnosis of 21 HLH patients and 50 untreated children with active juvenile dermatomyositis were evaluated by competitive enzyme immunoassay. HLH patients had higher levels of neopterin than the control group. Furthermore, neopterin significantly correlated with ferritin, suggesting a possible pathogenic link. Moreover, a cut-off of 38.9 nmol/L was derived by a ROC curve with a 70% sensitivity, and 95% specificity in diagnosing HLH, thus suggesting that neopterin levels could be an accurate marker of the disease [37].

Considering that IFN-γ is rapidly catabolized, it may be difficult to use it as a biomarker, thus highlighting the assessment of IFN-γ-induced chemokines in studying this pathway. In an elegant study, sera levels of the IFN-γ-induced chemokines (C-X-C motif) ligand 9 (CXCL9), and CXCL10 were evaluated in 14 patients with active HLH. These chemokines were higher than those collected from patients with a non-active disease or following therapies. Furthermore, the correlations among IFN-γ, CXCL9, and CXCL10 and the laboratory features of HLH were evaluated, including neutrophil and platelet counts, ferritin, lactate dehydrogenase, and alanine transaminase levels. CXCL9 correlated with all studied laboratory parameters. IFN-γ and CXCL10 correlated with all the parameters except for platelet counts for IFN-γ, and ferritin levels for CXCL10 [38]. In a further study, IFN-γ and IFN-γ-induced chemokines, CXCL9, CXCL10, and CXCL11, were studied using ELISA. Sera samples of 39 active and untreated AOSD patients, 30 rheumatoid arthritis patients, and 28 healthy controls were collected. IFN-γ, CXCL9, CXCL10, and CXCL11 were higher in AOSD patients when compared with RA patients or healthy controls. Furthermore, CXCL9, CXCL10, and CXCL11 were significantly higher in AOSD patients with MAS than those without it. In addition, these chemokines correlated with inflammatory markers and systemic scores. Notably, a decrease of these chemokines except for IFN-γ was observed after the reduction of disease activity during the follow-up. Finally, on immunohistochemistry, more inflammatory cells expressing CXCL10 were observed in skin biopsy samples from AOSD patients than in healthy controls [30].

### 4.3. Pathogenic Implications and In Vivo Observations 

The importance of IFN-γ in the pathogenesis of both primary and secondary HLH would be enhanced by the data obtained in experimental models. In fact, IFN-γ may be suggested as a pivotal mediator in murine models of HLH [39,40]. In this setting, the first experimental model of HLH was provided in perforin-deficient mice infected by lymphocytic choriomeningitis virus (LMCV). After this infection, the mice manifested the typical features of HLH, including fever, pancytopenia, and hypofibrinogenemia, associated with evidence of tissue hemophagocytosis. Furthermore, in this model, a marked increase of pro-inflammatory cytokines was shown with a remarkable quantity of IFN-γ. The latter was related due to a persistent antigen presentation and an increase in the antigen responsiveness of cytotoxic T cells [39]. Subsequently, in Rab 27a-deficient (Rab27a–/–) mice, it was also shown that infection with LCMV led to HLH [40]. Interestingly, in both these models, the administration of an IFN-γ blocking agent had a therapeutic effect [39,40]. In fact, the authors described how this treatment improved survival and led to an improvement of haematological and histopathological features in these mice. Indeed, the inhibition of IFN-γ increased blood cell counts. A significant reduction of triglyceride and ferritin levels was also observed over time in these experimental models. Furthermore, following IFN-γ inhibition, complete normalization of the histopathological features of the spleen was described in these models. The authors also noted a reduction of macrophage activation, as evidenced by the reduction of haemophagocytosis in the liver of both murine models [40].

In another study, it was shown that experimental HLH could be induced by repeated stimulation of TLR9 [34]. The authors also tested if IFN-γ could be required for the induction of HLH. Compared with WT mice, the IFN-γ–/–mice did not develop anaemia, thrombocytopenia, or hepatic inflammation, and these mice preserved the splenic structure. However, some features could not be dependent on IFN-γ, since leukopenia and hyperferritinemia were observed in both the WT and the IFN-γ –/– mice. Furthermore, the authors described the protective role of IL-10 in this setting, showing that the inhibition of its signal and/or the IL-10 receptor led to the development of hemophagocytosis. These data could reinforce the idea that IL-10 may also contribute by modulating both the variability and severity of this disease [34]. These findings were investigated in a later work in which IFN-γ-deficient mice underwent stimulation with a TLR9 agonist, IFN-γ, or a combination of both [35]. Following singular and repeated stimulation with a TLR9 agonist or IFN-γ, HLH features were not developed. However, mice treated with both a TLR9 agonist and IFN-γ reproduced the main features of HLH, developing cytopenias, hepatitis, and hepatosplenomegaly. On these bases, the authors suggested that TLR9- and IFN-γ-dependent signals could synergize in enhancing the myeloid progenitor function and inducing myelopoiesis. Thus, in this study, TLR9-driven signals would potentiate the effects of IFN-γ, leading to the development of HLH [35]. In a subsequent study, HLH in WT, transgenic, and cytokine-inhibited mice was assessed following stimulation with an IL-10 receptor-blocking antibody and a TLR9 agonist. Interestingly, fulminant HLH and hemophagocytosis developed independently of the presence of IFN-γ, whereas anaemia and dyserythropoiesis did not suggest an IFN-γ dependence [31]. IFN-γ dependent anaemia during HLH was also confirmed and detailed [41]. In fact, it was shown that IFN-γ could induce cytopenia and hemophagocytosis. The latter may have derived from the direct action of IFN-γ on macrophages in vivo, altering endocytosis and consequently leading to severe anaemia, the so-called consumptive anaemia of inflammation [41]. Other processes involved in HLH-associated anaemia could be blood loss, haemolysis, and decreased bone marrow output [41]. In addition, the IFN-γ-induced chemokines CXCL9 and CXCL10 were identified as possible biomarkers to be correlated with disease parameters including thrombocytopenia, hyperferritinemia, and lymphopenia [38]. These results provided the rationale for studying these IFN-γ-induced chemokines as possible predictors of HLH occurrence in humans, as previously mentioned [38]. 

In addition, some authors used a murine model of HLH induced by the administration of a TLR9 ligand in IL-6 transgenic mice to study the pathogenic mechanisms of the disease [36]. These mice, when injected with TLR ligands, may develop this condition by mimicking an acute infection on a background of high levels of IL-6 [36]. This experimental approach would more closely resemble what occurs in AOSD and its juvenile counterpart, an infectious trigger on an inflammatory background leading to MAS occurrence [36]. In addition, these IL-6 transgenic mice, following the administration of a TLR9 agonist, were associated with reduced survival, low neutrophils and platelet counts, and high levels of ferritin, LDH, and pro-inflammatory cytokines. In this experimental model, it was observed that IFN-γ and the IFN-γ-induced chemokines CXCL9 and CXCL10, were significantly increased in the liver, spleen, and plasma of the IL-6 transgenic mice, as compared to the WT mice. Furthermore, IFN-γ inhibition significantly decreased circulating levels of CXCL9, CXCL10, IL-1β, IL-6, TNF, and ferritin. Thus, a complex interplay between IL-6 and IFN-γ could be suggested in generating HLH [36].

### 4.4. Therapeutic Strategies

As previously discussed, experimental mouse models and ex vivo observations provide the rationale behind the use of IFN-γ inhibiting strategies for the treatment of HLH on account of the importance of the underlying IFN-γ-associated pathogenetic mechanisms of the disease [31,34,35,36,38,39,40,41].

Emapalumab is a fully human monoclonal antibody that neutralises both free- and receptor-bound IFN-γ by inhibiting receptor dimerization and the transduction of the signalling pathway of this molecule [42]. The efficacy of emapalumab was recently assessed in a clinical trial enrolling thirty-four patients aged between 0–18 years with a diagnosis of primary HLH, some were previously treated, while others were untreated. As main endpoints, the overall response was codified into patients with a complete response (defined absence of fever, cytopenia, hyperferritinemia, coagulopathy, neurological manifestations, increase of soluble CD25, and a normal spleen size), a partial response (three or more abnormalities that met the criteria for a complete response), or an improvement larger than 50% from baseline in at least three abnormalities associated with HLH. Twenty-six patients completed the eight-week treatment study. The percentage of previously treated patients with a response as assessed by the pre-defined parameters was 63%, while for the whole population of patients it was 65%. Of the previously treated patients, 26% achieved a complete response, 30% a partial response, 7% had improvement of HLH features, and 37% had no response. In the untreated patients, 43% achieved a partial response, 28.5% an improvement, and 28.5% no response. In this study, the authors also assessed CXCL9, which significantly decreased following the administration of emapalumab. Interestingly, low CXCL9 levels were associated with the clinical response during this clinical trial, suggesting possible predictors of efficacy following the administration of this drug [43].

In addition, a case report of a patient with refractory Epstein–Barr virus-associated HLH treated with emapalumab was recently described, with the resolution of all clinical symptoms and an improvement of laboratory markers of the disease [44]. Although IFN-γ inhibition would commonly be employed as a bridge to allogeneic stem cell transplantation, the successful use of emapalumab was also reported after transplant rejection in three relapsed primary HLH patients [45,46]. Finally, despite emapalumab being licensed for the treatment of primary HLH, several ongoing studies are assessing its use in the additional clinical settings of secondary HLH to systemic juvenile idiopathic arthritis (SJIA), and occurrence in adult ages (NCT03985423, NCT03311854).

## 5. IFN III

The third class of IFNs is composed of IFN-λ1, -λ2, -λ3, and -λ4 [16,47]. These are produced by most cell types, mainly from plasmacytoid dendritic cells following either viral or bacterial infection. Type III IFNs bind to the type III IFN receptor (IFNLR), preferentially expressed on certain myeloid cell types and epithelial cells of the respiratory, gastrointestinal, and reproductive tracts. This expression pattern is associated with local viral control at the site of entry. Furthermore, type III IFNs activate similar signalling pathways and partly induce the same genes as type I IFNs, resulting in a potent antiviral response [48]. The pathogenic role of type III IFNs in AOSD and MAS has yet to be defined.

## 6. Discussion and Appraisal of Literature

During AOSD, the difficult clinical scenario of MAS makes it difficult to manage patients, since genetic background, pro-inflammatory milieu, and triggers are mixed with a high mortality rate [5,6]. Thus, a growing body of studies has focused on investigating new therapeutic targets to better manage these patients [11,49]. Although IL-1 and IL-6 inhibiting agents were shown to be efficacious in AOSD [50,51], findings from clinical trials of canakinumab and tocilizumab on SJIA suggested that these therapies could not fully abrogate the risk of MAS development, even if the disease could be well controlled [52,53]. Consequently, these data suggest that additional pathogenic mechanisms could be implicated in MAS occurrence and together with the preclinical data, provided the rationale for IFN-γ inhibition in this field [31,34,35,36,38,39,40,41]. Thus, IFNs could be implicated in the development of this life-threatening complication during AOSD, as shown in Figure 1. In fact, Locatelli F et al. demonstrated the efficacy of emapalumab in children with primary HLH [43], which could be considered a genetic model of cytokine storm syndrome [54]. These clinical results may further confirm the pathogenic role of IFN-γ. It could also be possible to postulate the efficacy of emapalumab on cytokine storm syndromes from other aetiologies, including inflammatory or iatrogenic, and in adult ages. However, the data mined from children to adults with HLH would be limited by the presence of comorbidities, which may contribute to a higher rate of mortality in adulthood (almost 40%) [55,56]. In fact, patients with cytokine storm syndrome and comorbidities may be at high risk of poor prognosis, less able to tolerate medical procedures, and less responsive to any treatment, as recently shown in severe COVID-19 cases [57,58]. 

In addition, considering the poor prognosis of MAS occurring in AOSD, one crucial point would be a more accurate estimation of the subsequent clinical response. In this context, IFN-γ-induced chemokines were correlated with markers of MAS disease severity and clinical response to emapalumab [43,44], thus suggesting possible predictors of clinical response to treatment. Furthermore, IFN-γ-induced chemokines could be considered as mechanistic biomarkers, better reproducing the ongoing pathogenic mechanisms in MAS during AOSD and possibly more accurately reflecting the manipulated signalling pathways. In this context, specific HLH features such as anaemia and thrombocytopenia would be more correlated to IFN-γ [31]. In the heterogenous scenario of these patients, some clinical features should be considered as possible predictors of clinical response to IFN-γ inhibition when more relevant than others. Looking at new therapeutic strategies targeting IFN-γ, the possible role of JAK inhibitors was proposed in animal models of HLH as a further therapeutic option in these patients [59,60]. By the modulation of IFN-γ and other cytokines, the JAK1/2 inhibitor ruxolitinib reduced immune cell proliferation and activation, and reversed organ pro-inflammatory damage on experimental models of HLH [59,60]. Since they were concomitantly affecting different proinflammatory pathways, these drugs could simultaneously target IFN-γ and other pathogenic mechanisms of MAS during AOSD, possibly allowing for better management of cytokine storm syndrome in these patients [61]. On these bases, recent evidence has shown ruxolitinib may be considered for patients with secondary HLH with contraindications to glucocorticoids, with a good clinical response [62,63,64].

Finally, it must be pointed out that HLH could be also observed in patients with severe combined immunodeficiency lacking the main pathogenic effectors of the disease, T- and NK-cells. In these patients with severe combined immunodeficiency, the aberrant activation of macrophages and the subsequent cytokine storm syndrome may occur despite the complete absence of lymphocytes [65]. Furthermore, IFNGR1 deficiency is a rare immune deficiency characterized by selective susceptibility to mycobacterial disease due to IFNGR1 gene mutations [66]. Complete autosomal recessive IFNGR1 deficiency is characterized by the early onset of disseminated life-threatening infections from low-virulent mycobacteria, lack of response to IFN-γ cytokine replacement therapy, and high mortality [67]. A hematopoietic stem cell transplant is the only curative therapy available for these patients. Taking these observations together in the context of HLH, early identification of these patients would be needed to avoid unnecessary exposure to IFN-γ inhibition during cytokine storm syndrome.

## 7. Conclusions

In conclusion, IFNs are signalling molecules that mediate a variety of biological functions from defence against viral infections to antitumor and immunomodulatory effects. Preclinical and clinical observations suggest that these molecules could promote the hyperinflammatory response in MAS during AOSD, although additional evidence is needed to fully elucidate this topic. Finally, the positive results of inhibiting IFN-γ in primary HLH may provide a solid rationale to arrange further clinical studies, paving the way towards new therapeutic targets and reducing the high mortality rate in MAS during AOSD.

## Figures and Tables

**Figure 1 jcm-10-01164-f001:**
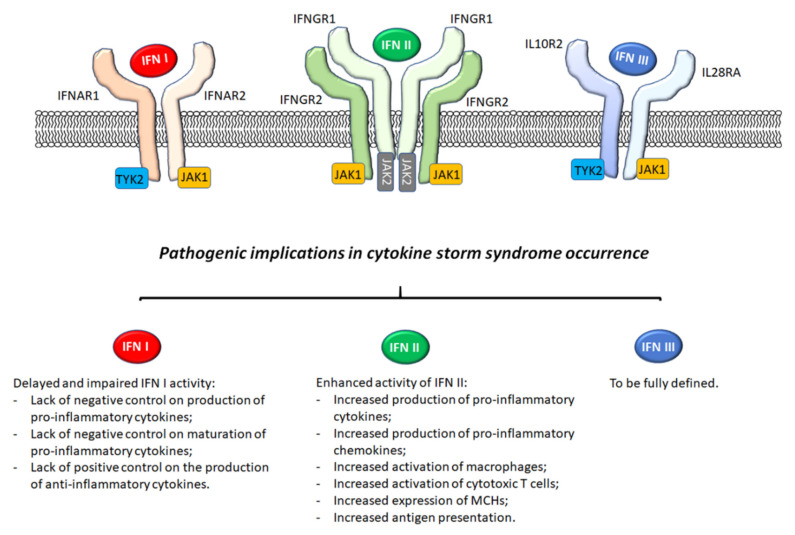
Pathogenic implications of IFNs in a cytokine storm syndrome occurrence. Type I IFNs bind to the IFNAR complex, consisting of two different chains, IFNAR1 and IFNAR2. Type II IFN activates the IFNGR, which is composed of two different chains, IFNGR1 and IFNGR2, and type III IFNs signal through a receptor complex made up of IL28RA and IL10R2. The impairment of the functions of type I IFNs or its delayed response may be implicated in the development of a cytokine storm syndrome lacking the negative control of production and maturation of pro-inflammatory cytokines as well as lacking the positive control on the production of anti-inflammatory cytokines. The enhanced activity of IFN II results in occurrences of cytokine storm syndrome via increased production of pro-inflammatory cytokines and chemokines and the increased activation of macrophages and cytotoxic T cells. The role of IFN III in this context has yet to be fully defined. Abbreviations: IFN: Interferon; IFNAR: interferon-alpha/beta receptor; IFNGR: interferon-gamma receptor; IL28RA: interleukin 28 receptor, alpha subunit; IL10R2: interleukin 10 receptor 2; TYK2: tyrosine kinase 2; JAK 1: Janus kinase 1; JAK 2: Janus kinase 2.

## Data Availability

Not applicable.

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
