# Peer review of "The Pathogenic Role of Interferons in the Hyperinflammatory Response on Adult-Onset Still’s Disease and Macrophage Activation Syndrome: Paving the Way towards New Therapeutic Targets"

_jcm, 2021, doi:10.3390/jcm10061164_

Round 1

Reviewer 1 Report

Di Cola et al. describe the roles of interferons in the context of MAS and HLH. The review is well-written, very informative and of high relevance. Some minor points should be adressed:  

  • „The latter is characterised by continuous high fever, extreme hyperferritinemia, pancytopenia, and histopathological evidence of hemophagocytosis by activated macrophages, typically in bone marrow [5,6].“ An important point is also organomegaly such as splenomegaly. However, hemophagocytosis is not necessary for diagnosis as it is often not present in initial bone marrow biopsies (see PMID 30992265).
  • Please replace „On this experimental model“ with „In this experimental model“
  • Ruxolitinib was also shown effective in secondary HLH

Author Response

Reviewer 1

Di Cola et al. describe the roles of interferons in the context of MAS and HLH. The review is well-written, very informative and of high relevance. Some minor points should be adressed: 

We thank the Reviewer for the interest in our work, we revised the suggested points.

„The latter is characterised by continuous high fever, extreme hyperferritinemia, pancytopenia, and histopathological evidence of hemophagocytosis by activated macrophages, typically in bone marrow [5,6].“ An important point is also organomegaly such as splenomegaly. However, hemophagocytosis is not necessary for diagnosis as it is often not present in initial bone marrow biopsies (see PMID 30992265).

As suggested, we added these observations in the introduction (lines 43-46) and the indicated reference (now reference 4).

Please replace „On this experimental model“ with „In this experimental model“

We replaced “on” with “in”.

Ruxolitinib was also shown effective in secondary HLH

As suggested, we reported the data about the efficacy of ruxolitinib in secondary HLH. Recent evidence has shown ruxolitinib may be considered for patients with secondary HLH with contraindications to glucocorticoid with a good clinical response [Wang J, Zhang R, Wu X, Li F, Yang H, Liu L, Guo H, Zhang X, Mai H, Li H, Wang Z. Ruxolitinib-combined doxorubicin-etoposide-methylprednisolone regimen as a salvage therapy for refractory/relapsed haemophagocytic lymphohistiocytosis: a single-arm, multicentre, phase 2 trial. Br J Haematol. 2021 Feb 9. doi: 10.1111/bjh.17331. Epub ahead of print. PMID: 33559893; Zhang Q, Wei A, Ma HH, Zhang L, Lian HY, Wang D, Zhao YZ, Cui L, Li WJ, Yang Y, Wang TY, Li ZG, Zhang R. A pilot study of ruxolitinib as a front-line therapy for 12 children with secondary hemophagocytic lymphohistiocytosis. Haematologica. 2020 Jul 30. doi: 10.3324/haematol.2020.253781. Epub ahead of print. PMID: 32732367; Hansen S, Alduaij W, Biggs CM, Belga S, Luecke K, Merkeley H, Chen LYC. Ruxolitinib as adjunctive therapy for secondary hemophagocytic lymphohistiocytosis: A case series. Eur J Haematol. 2021 Feb 1. doi: 10.1111/ejh.13593. Epub ahead of print. PMID: 33523540] (lines 359-360 and references 63-65).

Reviewer 2 Report

Di Cola et al. has extensively reviewed the pathogeneis of hyperferritinemic hyperinflammatory disorders, in particular MAS/sHLH, the role of interferon, and potential targets for intervention in these disorders.

The review article is well written, literature was robustly reviewed, and conclusions have been made with strong scientific rationale. The illustration describing the pathogenic implications of IFN in CRS is very helpful. 

I have only one comment- the authors should describe the IFNGR deficiency, and their genetic non responsiveness to anti-IFNG therapy. This has been reported in recent literature about how these disorders can use an alternate pathway, that are IFNg independent, driving the cytokine storm and hyperinflamamtion. These patients need to idenfied early in treatment, potentially use other alternate agents to achieve better disease control. 

Author Response

Reviewer 2

Di Cola et al. has extensively reviewed the pathogeneis of hyperferritinemic hyperinflammatory disorders, in particular MAS/sHLH, the role of interferon, and potential targets for intervention in these disorders. The review article is well written, literature was robustly reviewed, and conclusions have been made with strong scientific rationale. The illustration describing the pathogenic implications of IFN in CRS is very helpful.

We thank the Reviewer for the interest in our work.

I have only one comment- the authors should describe the IFNGR deficiency, and their genetic non responsiveness to anti-IFNG therapy. This has been reported in recent literature about how these disorders can use an alternate pathway, that are IFNg independent, driving the cytokine storm and hyperinflamamtion. These patients need to idenfied early in treatment, potentially use other alternate agents to achieve better disease control.

This is an interesting point raised by the Reviewer. In fact, HLH could be also observed in patients with severe combined immunodeficiency although lacking main pathogenic effectors of the disease, T- and NK-cells. In these patients with severe combined immunodeficiency, the aberrant activation of macrophages and the subsequent cytokine storm syndrome may occur despite the complete absence of lymphocytes [Bode SF, Ammann S, Al-Herz W, Bataneant M, Dvorak CC, Gehring S, et al. The syndrome of hemophagocytic lymphohistiocytosis in primary immunodeficiencies: implications for differential diagnosis and pathogenesis. Haematologica (2015) 100:978–88.  10.3324/haematol.2014.121608]. In this context, the IFNGR1 deficiency is a rare immune deficiency characterized by selective susceptibility to mycobacterial disease due to IFNGR1 gene mutations [Dorman SE, Picard C, Lammas D, Heyne K, van Dissel JT, Baretto R, Rosenzweig SD, Newport M, Levin M, Roesler J, Kumararatne D, Casanova JL, Holland SM. Clinical features of dominant and recessive interferon gamma receptor 1 deficiencies. Lancet. 2004 Dec 11-17;364(9451):2113-21. doi: 10.1016/S0140-6736(04)17552-1. PMID: 15589309]. Complete autosomal recessive IFNGR1 deficiency is characterized by early onset of disseminated life-threatening infections by low-virulent mycobacteria, lack of response to IFN-γ cytokine replacement therapy, and high mortality [Olbrich P, Martínez-Saavedra MT, Perez-Hurtado JM, et al. Diagnostic and therapeutic challenges in a child with complete interferon-gamma receptor 1 deficiency. Pediatr Blood Cancer. 2015;62:2036-2039;Gutierrez MJ, Kalra N, Horwitz A, Nino G. Novel Mutation of Interferon-γ Receptor 1 Gene Presenting as Early Life Mycobacterial Bronchial Disease. J Investig Med High Impact Case Rep. 2016 Nov 8;4(4):2324709616675463. doi: 10.1177/2324709616675463. PMID: 27868075; PMCID: PMC5103323]. To date hematopoietic stem cell transplant is the only curative therapy available for these patients. These observations would suggest an early identification of these patients to avoid the unnecessary exposure to IFN-γ inhibition (lines 362-374 and references 66-68).
